# Global Warming Mitigating Role of Wood Products from Washington State's Private Forests

**Indroneil Ganguly [1],\*, Francesca Pierobon [1] and Edie Sonne Hall [2]**

[1]  School of Environmental and Forest Sciences, University of Washington, Seattle, WA 98195, USA;
   pierobon@uw.edu

[2]  Three Trees Consulting, LLC, Madrid, NY 13660, USA; edieshall@outlook.com

\*  Correspondence: indro@uw.edu; Tel.: +1-206-685-8311

**Abstract:** Similar to standing trees in the forests, wood products play an important role in enhancing the global sequestered carbon pool, by retaining the atmospheric carbon in a sequestered form for the duration of the functional life of the wood products. This study uses a temporal radiative forcing analysis along with the functional half-life of different wood products to evaluate the impacts of wood products on global warming, including carbon storage and life cycle greenhouse gas production/extraction emissions. The methodology is applied to Washington State's aboveground biomass and timber harvest data, and to the State's comprehensive wood products mix. A moderate harvest rate simulation within Washington Biomass Calculator is used to estimate state harvest level, and statewide wood products manufacturing data is used for developing wood product mix estimates. Using this method, we estimate that the temporal carbon storage leads to a global warming mitigation benefit equivalent to 4.3 million $tCO_{2eq}$. Even after factoring in the greenhouse gas emissions associated with the harvest operations and wood products manufacturing processes, within the temporal model, the results show a net beneficial impact of approximately 1.7 million $tCO_{2eq}$, on an annual basis. It can further be noted that Washington State's annual biomass growth in its private forests exceeds its annual harvest, by a significant margin. This net yearly accumulation of biomass in the State's private forests leads to additional global warming mitigation benefits equivalent to 7.4 million $tCO_{2eq}$. Based on these results, we conclude that Washington's private forestry industry is a net global warming mitigator for the State, equivalent to 12% of the State's greenhouse gas emissions in 2015.

**Keywords:** forest products; global warming potential (GWP); industrial ecosystem; environmental assessment; life cycle assessment (LCA); carbon sequestration

---

## 1. Introduction

Mitigating climate change requires deep cuts in greenhouse gas (GHG) emissions by incentivizing the use of less intensive products worldwide [1]. The forest sector can help reduce atmospheric carbon dioxide ($CO_2$) by sequestering and storing carbon in trees and wood products and producing energy and materials that use relatively less fossil fuel compared to functional alternatives [2]. Understanding the role of forests managed for wood production (working forests) in mitigating global warming requires a comprehensive understanding of all the ecological components (i.e., carbon sequestration, GHG release through decay and decomposition), forest management activities (i.e., harvest, thinning activities, etc.), and wood products used on a long term temporal scale.

While forest management strategies may deliver carbon sequestration by increasing the standing biomass in the forest, the forest product pool also provides significant carbon storage benefits. By retaining the atmospheric carbon in a sequestered form for their functional lifetimes, wood

products play an important role in the global warming mitigation potential of the forestry sector [3–6]. Products made out of sustainably sourced wood displace other fossil intensive substitute products, such as concrete and steel [7–14]. Assessing the biogenic carbon stored in the form of biomass in the standing forests and wood products and estimating the resultant net carbon sequestration, on a temporal scale, in both pools combined, is essential to quantify the global warming mitigation potential of the forestry sector.

Wood products keep biogenic carbon sequestered from the atmosphere, for the duration of their functional life. This sequestration depends upon both the product's lifetime and its end-of-life fate. The longer the product is used, the longer the carbon stays in its sequestered form [15]. Additionally, once the product has reached its end of life, different disposal activities will affect carbon sequestration and climate impacts. For example, the product may be burned in place of fossil fuels, which would release the carbon but provide a substitution benefit for the displaced fossil fuels [16–18]. Alternatively, the product could be sent to a landfill, further increasing the duration of carbon sequestration, though emissions from decay may be in the form of methane ($CH_4$) due to anaerobic conditions [19].

Life Cycle Assessment (LCA) is the internationally recognized method used to evaluate the environmental impact of products and processes. The emphasis of LCA has always been on the emissions associated with the production, transportation, and disposal. Moreover, though we define LCA indicators with respect to a time horizon, e.g., a 100-year global warming impact, there is little consensus on how to factor in emissions happening at various time periods [20]. Traditionally, LCA relies on restricted steady-state models, and this reliance is considered to be a significant limitation, especially for the wood products and forestry sector [21–24].

Methodologies to calculate the effect of carbon storage and delayed emissions on global warming have been the object of study at the international level. Methods to account for the impact of carbon storage on global warming include the Moura-Costa and the Lashof accounting methods [25,26]. The Moura-Costa and the Lashof accounting methods have been discussed for the Kyoto Protocol GHG inventories as alternatives to the annual stock change method of the Intergovernmental Panel on Climate Change (IPCC) Good Practice Guidance for Land use, land-use change, and forestry [27] and have been lately proposed for LCA-related applications [25]. Based on those methods, the Publicly Available Specification (PAS) 2050:2008 introduced a simplified methodology to quantify the carbon storage that discounts carbon storage from the total emissions for the life of the product through a discount coefficient [28,29]. Currently, carbon accounting standards, such as the International Organization for Standardization (ISO) 14067, the GHG Protocol and the revised PAS 2050:2011, do not require that any credit is given to temporary storage in the base calculation [30–32]. Nevertheless, these standards allow a supplementary figure to be calculated that does include temporal aspects to be reported separately [25].

The Moura-Costa method evaluates the carbon storage by considering the number of years $CO_2$ is kept out of the atmosphere and converts it in terms of impact on global warming. The effect on global warming is then evaluated by estimating the cumulative radiative forcing (RF) integrated over a given time horizon caused by a pulse emission of one $tCO_2$. According to this calculation, storing one $tCO_2$ for 48 years is equivalent to avoiding the impact of one $tCO_2$ for 100 years [24,25]. Unlike the Moura-Costa method, the Lashof accounting evaluates carbon storage as a delayed emission. Based on this method, storing carbon for 48 years is equivalent to delaying a $CO_2$ emission until year 48. The impact of the delayed emission is evaluated through the cumulative RF of a pulse emission of one $tCO_2$ integrated from year 48 until the end of the time horizon, e.g., 100 years. Thus, the portion of the area of the curve beyond the 100-year time horizon, from year 100 until year 148, corresponds to the benefit of carbon storage [25].

To account for time in LCA, Levasseur et al. (2010) proposed a dynamic LCA approach [24] and applied it first to the U.S. Environmental Protection Agency LCA on renewable fuels [33]. The comparison of the results obtained using traditional and dynamic LCA showed that the difference can be significant, and that dynamic LCA leads to consistent results for global warming impact

assessment. The dynamic LCA approach was then applied to a temporary carbon sequestration project through afforestation, and the results were compared with those obtained using the Moura-Costa and the Lashof accounting methods [34]. The comparison showed that dynamic LCA is a more flexible approach that allows the consideration of every life cycle stage of the project and gives the opportunity to test the sensitivity of the results to the choice of different time horizons.

Other methodologies have used RF as a climate metric and have attempted to estimate the impact on global warming by averaging the amount of carbon that is predicted to remain in use in a wood product over 100 years, under the rationale that averaging roughly estimates the cumulative RF associated with the Bern Model [19]. This method is called the 100-year average methodology by the U.S. Forest Service and is also referred to as the California Forest Project Protocol [35,36].

Different approaches for estimating temporary carbon storage were compared in Levasseur et al. (2013) [37]. The approaches included a traditional LCA with and without consideration of biogenic carbon, the PAS 2050 and International Reference Life Cycle Data System Handbook methods, and a dynamic LCA approach. The study highlighted that the dynamic LCA approach allows for a consistent assessment of the impact, through time, of all GHG emissions and sequestration [37]. Similarly, a recent study summarized 15 different climate change impact assessment methods and presented a comparison of the results obtained when applying these methods to three simplified bioenergy systems [38]. The authors highlighted the importance of interpreting the results based on an understanding that different methods focus on different aspects of climate change and represent different time preferences.

*Study Objectives*

The objective of this study is to develop a comprehensive evaluation of the climate change benefits associated with sequestered carbon in wood products over the respective lifetimes of the products. To achieve this objective, the study factors in:

(i)   $CO_2$ decay in the atmosphere using the Lashoff $CO_2$ accounting method.
(ii)  GHG emissions associated with harvesting and manufacturing wood products, using the example of current harvest levels and product mix in Washington State.
(iii) Functional life of the wood products using the U.S. Forest Service half-life assessment of various categories of wood products.
(iv)  Net carbon storage benefits of wood products, after factoring in all the production emissions associated with the wood product mix under consideration.

Accordingly, this study examines the temporal components of the wood product carbon pool in order to advance wood product LCA methodology, building upon previous research on RF analysis [21–25,37–41]. In this study, we expand current methodologies to account for the carbon storage within wood over the functional life of the wood products, by considering the fraction of carbon in primary wood products remaining in end uses up to 100 years. Additionally, we apply the method to the private forests of Washington State's aboveground biomass harvests and the corresponding empirical wood waste and wood products mix produced.

It may be noted that this study's focus is on the attributional analysis of wood products and does not address the substitution effect. Moreover, this study specifically estimates the impact of the existing practices. Though the proposed model can be used for developing what-if scenarios, it is outside the scope of this study.

## 2. Materials and Methods

The fundamental premise of the study is that the carbon that stays sequestered in wood products, during their functional life, is equivalent to avoiding the emission of a corresponding amount of $CO_2$ for that period. Upon harvest a significant proportion of the aboveground biomass ends up in wood products and the corresponding carbon stays in a sequestered form over the functional life of the wood product. Accordingly, the proposed method considers the global warming mitigation

potential of this avoided emissions on a temporal scale, factoring in the resultant product mix and their corresponding functional lifespans. The carbon storage in wood products is evaluated through the cumulative RF of a pulse emission of one $tCO_2$ integrated from the beginning of the time horizon until the last year of storage or 100 years, whichever is smaller [25]. The actual product lifetimes are used to calculate the global warming mitigation potential by considering the amount of sequestered carbon within a specific wood products biomass still in use at each year based on the product functional life [24,25]. All the GHG emissions associated with the harvest and production of wood and wood products are also factored into the model. On the emissions side, the GHGs emitted are also evaluated using a similar method, where the cumulative RF of a pulse emission of one $tCO_2$ is integrated over 100 years, independent of the actual functional life of the wood products. Within this paradigm, the net global warming mitigation potential of Washington's private forests is estimated using the multi-step approach described in the sections below.

The analysis is conducted using a 100-year time horizon, following the IPCC rules on GHG decay rates in the atmosphere and their corresponding RF. This study limits its scope to the functional first life of wood products and does not consider the reuse, recycle, and landfilling aspects of the wood products. The estimation method proposed in this paper addresses this gap in the literature by developing a comprehensive understanding of the global warming mitigation potential of stored carbon within wood products. This method estimates the current inventory and inventory change of wood products based on past harvest data and using U.S. Forest Service statistics on wood product end-use distribution and half-lives.

## 2.1. Assumptions

Carbon Neutrality: We established that the total yearly carbon sequestration of biomass through growth in private forests offsets the amount of harvested biomass from the corresponding forests before using the carbon neutrality assumption of biogenic carbon (detailed in Section 3.1 of this paper).

Wood Products Mix in Washington State: We used the data published by the Washington Mill Survey of 2016, by Washington Department of Natural Resource [42], as the representative product mix for wood harvested in Washington State. Though not all wood harvested within Washington State is processed within the State, we assumed that the State product manufacturing mix is a good representation for the biomass harvested within the State.

Forest Growth/Harvest Rate: We used the moderate scenario within the Washington Biomass Assessment simulation tool [43] to estimate the biomass growth and harvest rates within Washington's private forestlands. It may be noted that regular forest fires are factored in the net growth (i.e., growth minus mortality) and harvest rate models. This paper also assumes no land-use change scenario, i.e., no conversion of forestland for non-forestry uses or vice versa. It may be noted that this tool produces results that align with the recently published Washington's harvest and growth data [44].

Simplifying assumptions regarding the functional life of wood products: In this study, we assumed that wood products have a single life and no option of recycling. Adding the possibility of recycling would most likely increase the life of sequestered carbon in wood products; however, to simplify the modeling process, we opted to leave it out of our current study. We also did not consider the landfilling and corresponding methane recovery options for similar reasons.

## 2.2. Evaluation of the Total Harvest Biomass in Private Forests of Washington State

In Washington State, most of the private forestland is categorized as timberland. A recent report estimates that the total area of private timberland in the State is 3.7 million ha out of a total of 3.8 million ha of private forestland [44]. The report also states that the average annual biomass growth in Washington state is 8.2 $m^3$ $ha^{-1}$ with an annual mortality rate of 1.5 $m^3$ $ha^{-1}$. After factoring in the average annual removals (though harvest/thinning operations) of 5.3 $m^3$ $ha^{-1}$, the net average annual change in forest biomass is 1.4 $m^3$ $ha^{-1}$ on Washington's private land [44].

In this study, in-forest biomass change is assessed by expanding the Forest Inventory and Analysis data to the parcel level, using the Washington Forest Biomass Supply Assessment tool [43]. Forested plots are simulated using the appropriate Forest Vegetation Simulator variant to capture the variation in growth and yield in the state over a 20-year period (2010 to 2030). A harvest scenario was designed to mimic actual harvest levels from the state for the year 2015. The resultant model estimated the 2015 harvests of 10 million bone dry tons of total aboveground biomass harvest (7.2 million bone dry tons from private and 2.8 million bone dry tons from other forest types), which closely resembles the empirical harvest from the state during that year of 11.35 million bone dry tons [45]. Harvest targets are defined by county and ownership based on published Washington Department of Natural Resources harvest reports, which are used to constrain harvest activity up to the targeted volume in a specific area and for specific land ownership classes. The inventory for Washington produces 6085 unique forest class plots, of which 5998 forested. All owner information and parcel geometry for the Biomass Database are derived from the 2009 Washington State Parcel Database.

The Forest Vegetation Simulator results include the total aboveground and harvest volumes and the total residual biomass (expressed in terms of mass). Results are aggregated for five years; e.g., the 2015 biomass refers to the 5-year biomass from 2010 to 2015. We estimate aboveground live-tree and harvest biomass using specific gravity constants for each species to compute the dry weight of the tree [46]. The Forest Vegetation Simulator results are used to calculate annual carbon sequestration, by assuming a carbon content in the biomass of 50%. All references to weight are in oven-dry units. The Forest Vegetation Simulator results are stratified by landownership categories, forest ecosystem types, species, and location (parcel level). From the total database, the results for private forests (large and small) are extracted. The results are then grouped into five main economic areas. As reported in the Department of Natural Resources 2016 Washington Mill Survey, an economic area is determined by the similarity of economic activity in the forest products industry. Washington's main economic areas are: Puget Sound, Olympic Peninsula, Lower Columbia, Central Washington, and Inland Empire.

*2.3. Creation of a Wood Products Mix Scenario*

The primary production data from the Department of Natural Resources 2016 Washington Mill Survey is used to calculate the wood product mix [42]. This survey reports data about all primary log consuming operations, such as lumber, veneer and plywood, pulp, shake and shingle, log exports, post-pole-piling, and chip operations. For the log export, the same product mix used for domestic production is assumed for the study.

Based on the Department of Natural Resources data about primary production, the input and output flow for each wood product industry at the county level are calculated. Primary production includes lumber, roundwood chipping, pulp and board, veneer and plywood, and other products. While producing lumber, shakes, and plywood, the mills generate a large volume of mill residues, such as chips, bark, sawdust, and shavings. These mill residues are sold for pulp, fuel for bioenergy production (hog fuel), feedstock for board manufacturing, garden mulch, and livestock bedding. Less than 1% of mill residue generated by Washington mills is reported as unused waste.

Various sources of production data are used to create a wood products mix scenario, including different uses of the merchantable harvest from private forests in Washington State in 2015 [42]. Hog fuel, bark, and wood fuel are considered as hog fuel. Hog fuel and waste are excluded from the storage evaluation because their lifetime is <1 year.

*2.4. Assignment of Wood Product Functional Lifetimes*

The global warming mitigating potential of wood products is calculated by considering the functional lifetime of different wood products. We used the U.S. Department of Agriculture report which provides the data on the fraction of different primary wood products remaining in their respective functional use, up to 100 years after production (Table 6-A-2 in Hoover et al. (2014)) [19]. This report provides values for softwood lumber, hardwood lumber, softwood plywood, oriented strand board,

non-structural panels, miscellaneous products, and paper. To construct the primary product type decay tables, data were used on the disposition of each primary product to major end uses (e.g., percentage of product going to residential housing, non-residential housing, manufacturing (furniture), and exports) [19], with residential construction being a primary end-use of wood products in the U.S. [47]). In our analysis, we used the data for softwood lumber, softwood plywood, particleboard, and paper for modeling the statewide global warming potential of the wood products industry. We assume that there is a national market for primary products and the percentage of primary products going to each end-use is the same across the country. We also assume that primary products exported from the United States are used in the same way as domestic products. Based on Hoover et al. (2014), the half-life is 38 years for softwood lumber and plywood, and five years for paper. We also use the report to compare the RF results with the 100-year average fractions (Table 6-A-2 in Hoover et al. (2014)). Given the focus of this study is wood products, the carbon storage is assumed to start at the beginning of the evaluation period (time = 0), which is the year of harvest, until the end of the product life.

### 2.5. Calculation of Emission Profiles

The evaluation includes cradle-to-gate fossil emissions, including harvest, transportation, and production emissions. Greenhouse gas emission values are taken from the U.S. Life Cycle Inventory (LCI) database [48]. The U.S. LCI databases used for each wood product are reported in Table 1. Paper is assumed as a mix of freesheet, coated and uncoated, and mechanical, coated and uncoated, and the average emission value of the corresponding U.S. LCI databases is considered.

**Table 1.** US Life Cycle Inventory (LCI) databases used for the wood products greenhouse gas ($CO_2$ and $CH_4$) emission values.

| Wood Products | U.S. LCI Database |
|---|---|
| **Lumber** | Sawn lumber, softwood, rough, kiln-dried, at kiln, $m^3$/PNW_US |
| **Plywood** | Plywood, at plywood plant, US PNW/kg/US |
| **Paper** | Paper, freesheet, coated, average production, at mill/kg/RNA |
| | Paper, freesheet, uncoated, average production, at mill/kg/RNA |
| | Paper, mechanical, coated, average production, at mill/kg/RNA |
| | Paper, mechanical, uncoated, average production, at mill/kg/RNA |
| **Miscellaneous** | Particleboard, average, softwood, particleboard mill/$m^3$/RNA |

Based on the U.S. LCI data of wood products, $CO_2$ and $CH_4$ emissions constitute more than 99% of the GHG emissions from wood products production and biomass decay process. Hence, this study factors in the life cycle $CO_2$ and $CH_4$ emissions of all harvest and wood products manufacturing operations. The time horizon of evaluation of the emission profiles is 100 years. The production emissions are considered as pulse emissions released within one year of harvesting (time = 0). As has been identified in Section 2.1, we assume that all the biogenic carbon in wood products is emitted to the atmosphere immediately after its functional life.

### 2.6. Calculation of the Radiative Forcing for a Pulse of Greenhouse Gas

2.6.1. Decay Functions of Greenhouse Gases in the Atmosphere

The impact of GHG emissions on global warming is calculated through the RF concept, which is related to the relative abundance of that GHG in the atmosphere and its radiative efficiency (RE). The relative abundance of a GHG in the atmosphere is measured through the decay functions, which tell how long a unit-pulse of GHG will stay in the atmosphere once it gets released, due to the environmental

capacity to transform or remove it from the atmosphere. This environmental capacity depends on the GHG residence time in the atmosphere and its bulk concentration. The decay of a pulse of $CO_2$ with the time $t$ is based on the revised version of the Bern carbon cycle model and is given by:

$$C_{CO_2}(t) = a_0 + \sum_k a_k e^{-t/\tau_k}, \tag{1}$$

$a_0 = 0.2173$; $a_1 = 0.2240$; $a_2 = 0.2824$; $a_3 = 0.2763$; $\tau_1 = 394.4$ years; $\tau_2 = 36.54$ years; $\tau_3 = 4.304$ years.
　The decay of a pulse of GHG ($CO_2$ excluded) follows a first-order decay equation, given by:

$$C_{GHG_i}(t) = e^{-\frac{t}{\tau_i}}, \tag{2}$$

$\tau_i$ = lifetime of the *i*-th GHG. The values of the GHG lifetimes according to the 5th IPCC Report are $\tau_{CH4} = 12.4$ years and $\tau_{N2O} = 121$ years [49].

### 2.6.2. Calculation of the Radiative Efficiency

The radiative efficiency is the RF per unit mass increase in the atmospheric abundance of component $i$ and is calculated for a perturbation of one unit to the background concentration (Table 8.SM.1, Chapter 8, Supplementary Material, 5th IPCC Report [49]). According to IPCC, to convert the radiative efficiency values from parts per billion by volume (ppbv) to per kg, they must be multiplied by $(M_A/M_i)(10^9/T_M)$, where $M_A$ is the mean molecular weight of air (28.97 kg kmol$^{-1}$), $M_i$ is the molecular weight of species *i* and $T_M$ is the total mass of the atmosphere, $5.1352 \times 10^{18}$ kg [50,51]. The RE values per ppbv are reported in Table 8.A.1 of the IPCC 5th Report. The radiative efficiencies thus calculated are $1.76 \times 10^{-15}$ W m$^{-2}$ kg$^{-1}$ for $CO_2$ and $1.28 \times 10^{-13}$ W m$^{-2}$ kg$^{-1}$ for $CH_4$. We included indirect effects of methane on ozone and stratospheric water by increasing the RE of methane by 65%, consistent with AR5 [52].

### 2.6.3. Calculation of Radiative Forcing for a Pulse of Greenhouse Gas

The RF for a pulse of GHG is obtained by multiplying the decay function for a pulse of GHG by the radiative efficiency of the GHG for each time $t$.

$$RF_{GHG_i}(t) = RE_{GHG_i} \cdot C_{GHG_i}(t), \tag{3}$$

Emissions have a positive *RF*, while sequestered carbon dioxide has a negative *RF*. The number of equations is equal to the number of GHGs included in the study. In this study, two RFs are calculated, one for each of the considered greenhouse gases, $CO_2$ and $CH_4$.

### *2.7. Application of the Radiative Forcing Analysis to the Emission Profiles*

### 2.7.1. Calculation of the Emission Profile Vectors

To apply the RF analysis to the emission profiles, a vector for each emission/removal source $j$ and for each greenhouse gas $i$ is defined (emission profile vectors). In this study, three vectors are calculated: one for carbon storage, and two for manufacturing emissions (one for each of $CO_2$ and $CH_4$). For carbon storage, the components of the vector are the $CO_2$ values corresponding to the fractions of carbon in primary wood products remaining in end uses, from year 0 to year $n$, where n is the last year of the evaluation period (i.e., year 99).

$$E_{j,\,GHG_i}(t) = \left( e_j(t_1) e_j(t_2) \cdots e_j(t_{n-1}) e_j(t_n) \right), \tag{4}$$

For manufacturing emissions, the components of the vector are the lifecycle-based fossil GHG emissions from wood product manufacturing. Given that all emissions occur at the beginning of the evaluation period, $e_j(t_1) = e_j(t_2) = \cdots = e_j(t_{n-1}) = e_j(t_n)$. The dimension of all vectors is equal to the number of years of evaluation, i.e., 100.

2.7.2. Application of the Radiative Forcing to the Emission Profiles

By multiplying each emission profile vector by the RF of the corresponding greenhouse gas unit-pulse, the cumulative radiative forcing (CRF) for the source $j$ and $GHG_i$ is obtained:

$$CRF_{j,GHG_i} = E_{j,\,GHG_i}(t) \cdot RF_{GHG_i}(t) = \left(e_j(t_1)e_j(t_2)\cdots e_j(t_{n-1})e_j(t_n)\right) \cdot \begin{pmatrix} RF_{GHG_i}(t_1) \\ RF_{GHG_i}(t_2) \\ \cdots \\ RF_{GHG_i}(t_{n-1}) \\ RF_{GHG_i}(t_n) \end{pmatrix} \tag{5}$$

It should be noted that the radiative efficiencies are expressed in $W\,m^{-2}\,kg^{-1}$; therefore, the emission profiles should be expressed in kg of greenhouse gas. Summing up the cumulative radiative forcings of different GHGs for each GHG emission/removal source, the total cumulative radiative forcing is calculated for each of the two contributions: carbon storage and production emissions ($CO_2$ and $CH_4$):

$$CRF_j(t) = \sum_i CRF_{j,GHG_i}(t) \tag{6}$$

where, $CRF_j(t)$ represents the cumulative radiative forcing at any point of time $t$.

2.7.3. Calculation of the Net Cumulative Radiative Forcing

The net cumulative radiative forcing (NCRF) is calculated by summing the cumulative radiative forcings of each emission/removal source:

$$NCRF = \sum_j CRF_j(t), \tag{7}$$

where the net cumulative radiative forcing is a measure of the impact of the woody biomass system biogenic carbon emissions and sequestration on global warming.

2.7.4. Calculation of the Global Warming Potential Metrics

GWP is the time-integrated RF of a pulse emission of a GHG relative to that of a pulse emission of $CO_2$. The IPCC 5th assessment report has provided updated GWP values for all GHGs [49]. The GWP metric for wood products is calculated as the net cumulative radiative forcing of the product (calculated summing up cumulative radiative forcings of each emission source) at 100 years relative to the cumulative radiative forcing of $CO_2$ at that same time.

$$GWP = \frac{\sum_j CRF_j(t)}{\sum CRF_{CO_2}(t)}, \tag{8}$$

The calculations are performed using R v.3.2.4, a free software environment for statistical computing and graphics.

*2.8. Evaluation of the Total Global Warming Mitigation Potential of Washington State's Wood Products*

The global warming mitigation potential of each wood product is applied to the Washington state wood products mix. The detailed analysis focuses only on wood products harvested from private lands (i.e., not including state or tribal lands) in Washington state, which comprises about 73% of the total harvest in 2015. The impact of the different wood products is calculated over a 100-year time horizon with and without the fossil emissions associated with their production processes. The total GWP is calculated considering both the fossil emissions and the carbon storage of wood products.

## 3. Results

### *3.1. Forest Harvest in Washington State's Private Forests*

The Forest Vegetation Simulator model produces results stratified by landownership categories, forest ecosystem types, species, and location, at the parcel level. The forest harvest results are represented in Figure 1. The green areas and the pink areas represent the distribution of harvested biomass within Washington State. The red dots on the map indicate the quantity harvested in each of the counties in 2015, with bigger dots indicating larger harvest volumes. In 2015, Lewis County contributed the most significant volume of logs to in-state mills, followed by Grays Harbor and Cowlitz counties. In eastern Washington, the top timber-supplying counties to in-state mills were Stevens and Pend Oreille counties.

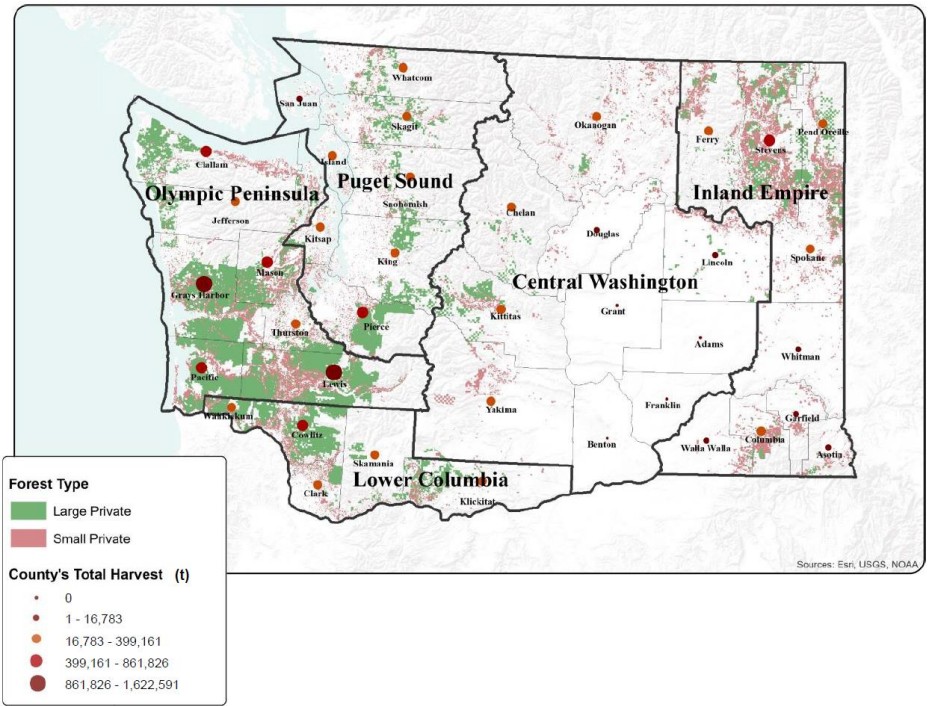

**Figure 1.** Total harvest in Washington state's private forest in 2015.

Total merchantable harvest values for 2015 in private forests (large and small) are reported in Table 2 by economic area as defined in the Department of Natural Resources 2016 Washington Mill Survey.

**Table 2.** Total merchantable harvest in private forests by economic area for 2015. All values are reported in bone-dry tons.

| Economic Area | Total Harvest (t) | Total Harvest, Large Private (t) | Total Harvest, Small Private (t) |
|---|---|---|---|
| Puget Sound | 1,329,670 | 984,999 | 344,670 |
| Olympic Peninsula | 3,760,725 | 2,871,708 | 889,019 |
| Lower Columbia | 1,085,829 | 848,913 | 236,916 |
| Central Washington | 222,756 | 106,814 | 115,942 |
| Inland Empire | 779,369 | 364,385 | 414,983 |
| Total Washington | 7,178,348 | 5,176,819 | 2,001,530 |

As shown in the table, more than half of the timber harvest processed by Washington mills comes from the Olympic Peninsula economic area, while about 19% comes from the Puget Sound, 15% from

Lower Columbia, 11% from Inland Empire, and the remaining 3% originates in Central Washington. Of the total harvest in Washington's private forests, about 72% comes from large private and 28% comes from small private forests.

Washington's forests are currently a net sink for carbon, as the growth of trees exceeds harvest and mortality overall. Based on our projections, between 2010 and 2030, growth volume will significantly exceed removals on private timberland at the state level. This result is demonstration of sustainable forest management assured by the state's forest practices rules in addition to third-party forest certification on 3.1 million hectares of timberlands [53–55]. Given the overall yearly biomass sequestration in Washington's private forests is greater than the harvests; we can assume the GWP neutrality of the biogenic carbon contained in the wood products.

### 3.2. Wood Products Mix

To get an accurate volume estimate of the end-use specific primary wood products produced in the State, it is essential to match the merchantable harvested biomass to the regional processing facilities. A comprehensive mass balance of all the merchantable biomass harvested in the state was completed to understand the interlinkages between the processing facilities in the State and the volume of waste. Accordingly, a flowchart with all the input and output flows for the production of different wood products is represented in Figure 2.

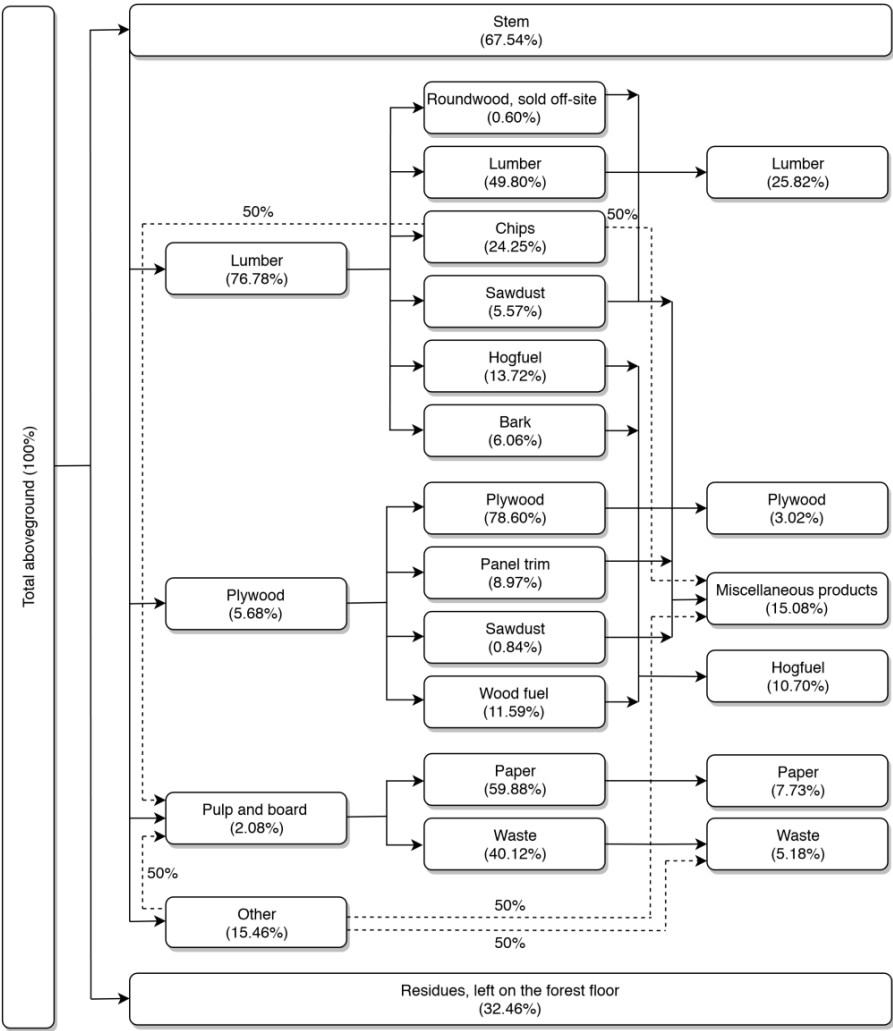

**Figure 2.** Input and output flows for the production of different wood products, on a dry mass basis.

Of the harvested tree, the stem represents about 67.54%, while residues (tops, branches, and foliage) represent about 32.46% of total biomass. Of the stem, about 76.78% goes to the lumber industry; 15.41% is used to produce wood chips for pulp, particleboard, and other uses; 5.68% goes to the veneer and plywood industry; 2.08% is used by the pulp and paper industry, and 0.05% is sent to the pole industry. We combined this data about primary production with data about the production processes in each wood product industry, which includes the inputs and outputs flows of different co-products.

During lumber production, several co-products are produced: lumber (49.80%), chips (24.25%), sawdust (5.57%), hog fuel (13.72%), bark (6.06%) and roundwood, sold off-site (0.60%). The plywood production leads to the production of plywood (78.60%), panel trim (8.97%), sawdust (0.84%), and wood fuel (11.59%). In paper production, about 59.88% of the biomass is converted to paper, and the remaining 40.12% is waste. Waste in paper production consists of rejects, different types of sludges and ashes in mills having on-site incineration treatment. Roundwood sold off-site refers to the small quantity of logs that leave the facility because the logs are not suitable for sawing [53]. Logs sold off-site may have a lower value (as in the case of a defective log) or higher value (a veneer log, for example). For the purpose of this study, they were considered as "miscellaneous" products.

Based on the sawmill survey data [53], it is assumed that 50% of the chips produced during lumber production and 50% of the chips produced in chip mills are used for paper production. Adding these contributions to the primary production, the overall paper production from Washington state's private forests is estimated to be 7.73% of the total aboveground harvested biomass. The trimmings and bark produced during lumber production and the wood fuel produced during plywood production are classified as hog-fuel, which accounts for 10.70% of total harvests. The waste generated from paper production is disposed of in the landfill, which contributes to 5.18% of the total harvests. Particleboards are produced from 50% of the chips produced during lumber production, 50% of the chips produced in chip mills, and from the sawdust and panel trim. Particleboards and the remaining contributions (i.e., roundwood discarded from lumber production and sold off-site) are classified as "miscellaneous" and contribute to 15.08% of the total biomass of the harvested tree. Excluding residues, hog fuel and the wood portion that ends up in landfills, the wood products mix for the overall Washington state and by economic area is calculated (Table 3).

**Table 3.** Wood products mix for the overall Washington state and by economic area from merchantable harvested biomass from private forests in 2015. All values are reported in bone-dry tons.

| Wood Products | Washington State (t) | Puget Sound (t) | Olympic Peninsula (t) | Lower Columbia (t) | Central Washington (t) | Inland Empire (t) |
|---|---|---|---|---|---|---|
| Lumber | 2,550,470 | 582,840 | 1,182,310 | 506,183 | - | 279,137 |
| Plywood | 319,284 | 39,506 | 225,242 | 54,536 | - | - |
| Paper | 919,360 | 117,632 | 555,001 | 73,809 | 66,693 | 106,225 |
| Miscellaneous | 1,713,116 | 273,546 | 923,484 | 192,747 | 111,378 | 211,961 |
| Total | 5,502,230 | 1,013,524 | 2,886,037 | 827,276 | 178,071 | 97,323 |

Washington's overall wood products mix is dominated by lumber, with 46% share of the overall pie, the second largest single category is pulp and paper with 17% share of the pie, plywood constitutes about 6%, and the remaining 31% is miscellaneous wood products. The results are combined with the total harvest values calculated using the Forest Vegetation Simulator.

*3.3. Global Warming Potential of Wood Products*

The $CO_{2eq}$ decay is calculated over a 100-year time horizon for the different wood products produced in 2015 with and without the fossil emissions associated with their production processes. In Figure 3, the negative y-axes region of the graphs represents the reduction in RF as a result of avoiding a $CO_2$ emission equivalent to the temporary biogenic carbon storage within the wood products. The positive y-axis region of these graphs represents the emissions associated with life cycle

GHG gases (fossil-fuel based $CO_2$ and $CH_4$), converted to $CO_{2eq}$ values using the radiative efficiency ratio of the GHGs under consideration, as a result of harvest and production activities associated with the corresponding wood products. Each of these graphs considers a 100-year timeframe, represented on the x-axis. The green area in the graph represents the avoided $CO_{2eq}$ decay from wood product carbon storage. The red area represents the $CO_{2eq}$ decay from wood product production emissions. Visually, for any product with a green region (integral of the area under the green curve) larger than its corresponding red region (integral of the area under the red curve), we can conclude its temporary carbon storage benefit outweighs its emission burdens, and the product is a net global warming mitigator on a 100-year scale.

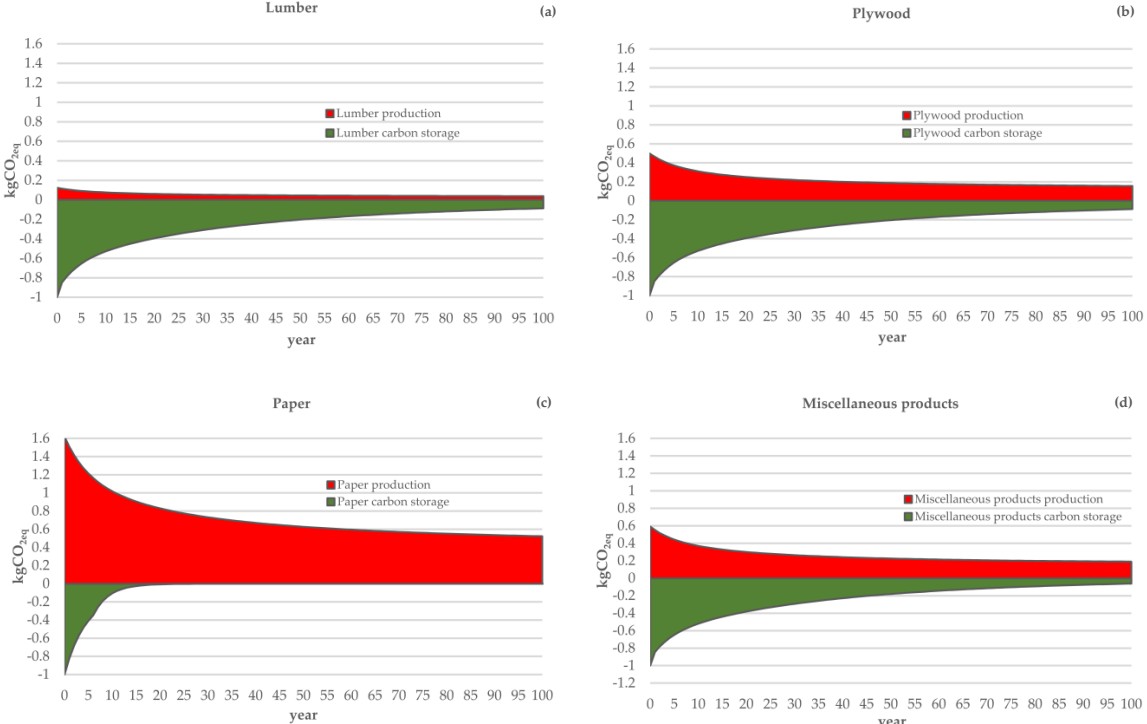

**Figure 3.** Comparison between the $CO_2$ decay from wood products manufacturing fossil fuel-based life cycle emissions (red), and avoided $CO_2$ decay from wood products carbon storage (green), for 1 kg of (**a**) softwood lumber; (**b**) softwood plywood; (**c**) paper products; (**d**) miscellaneous products (particle boards and roundwood from lumber production sold off-site). The time horizon of the evaluation is 100 years.

The results for lumber, plywood, paper, and other miscellaneous wood products, are represented in panels (a) to (d) of Figure 3. The results indicate that lumber, plywood, and other miscellaneous wood products have significant climate benefits, given the integral of the green area is larger than the red one. In our analysis, the emissions associated with paper production still outweigh the potential climate benefit of the carbon stored in paper, making paper products a net contributor to global warming. This is driven by two factors: the large quantity of emissions associated with producing paper and the relatively short lifetime of paper.

Given the half-life of $CO_2$ (the most abundant greenhouse gas) is much higher than most of the wood products, the net sequestration benefits of wood products are greater in the short-term than in the long-term. Accordingly, the net global warming mitigation value of any wood product would be higher on a shorter time range. For example, if we want to develop a 50-year global warming reduction plan, wood products would play a greater mitigating role, as compared to a 100-year plan. On the contrary, products like plywood, which is a net global warming mitigator on a 100-year scale, would end up net neutral on a 200-year scale. In this study, we used the widely adopted 100-year temporal

scale for our analysis. The results of net GWP over a 100-year timeframe, performed with and without including fossil emissions, are represented in Figure 4. Fossil emissions are already included in the Washington Greenhouse Gas Emission Reduction Limits Report [56]. The results of this study are reported without including fossil emission to allow comparison with the greenhouse gas emissions published in the report and avoid double-counting. The fossil emissions play an important role in the global warming potential of the wood products industry. Lumber has the most significant climate change benefit out of all forest product scenarios that we considered. This is primarily driven by the relatively low emissions associated with producing lumber. Paper products result in a positive GWP value (i.e., increasing climate change) when we consider emissions associated with their production. When we do not consider the emissions, paper products only provide a small climate benefit due to their short lifetime. Out of all products considered, paper products have the highest level of emissions associated with their production.

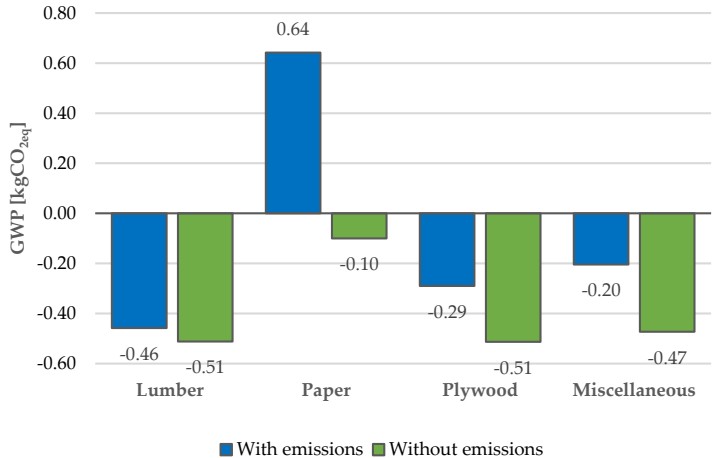

**Figure 4.** Global warming mitigating role of wood products from private forests in Washington state, calculated over a 100-year timeframe. Results include the scenario where fossil fuel-based life cycle emissions from wood products manufacturing are included (blue) and excluded (green).

These results indicate that the carbon stored in the biomass of wood products could further enhance the carbon sequestration capabilities of forests and reduce climate change impacts. Under some production scenarios, the emissions associated with production will outweigh the potential climate benefit of carbon storage in a product, such as in the case of paper.

This product-specific RF analysis also demonstrates the global warming benefit of storing carbon in long-lasting forest products. Effective carbon storage policies could, therefore, encourage extending product lifetimes, such as through reuse or refurbishment, to keep biogenic carbon in its sequestered form. Extending forest product lifetimes may even allow storing carbon for a longer timeframe compared to long-term carbon storage in the forest. Old-growth stands are subject to biotic agents of forest disturbance and have high mortality rates [57], which may prevent carbon from being stored for a long timeframe.

*3.4. Global Warming Mitigating Role of Wood Products in Washington State*

The GWP numbers for individual products are multiplied by the overall Washington wood products mix, to estimate the net environmental impact of the wood products industry (Figure 5). Based on the 2015 Washington State wood products' production data, our research shows that, overall, the wood products industry has a net global warming mitigation benefit.

Wood products produced from private forests in Washington State contribute to global warming mitigation of about 4.3 million tons of $CO_{2eq}$ in 2015 if production emissions are excluded. When production emissions are included, there is still a net benefit from just the wood products alone,

equivalent to about 1.7 million tons of $CO_{2eq}$ in 2015. All these benefits are underpinned by net carbon sequestration on Washington State lands.

In 2008, the Washington Legislature established greenhouse gas emission reduction goals, which include reducing overall greenhouse gas emissions in the state to 1990 levels by 2020, to 25% below 1990 levels by 2035 and 50% below 1990 levels or 70% below the state's expected emissions that year by 2050 [56]. According to the Department of Ecology's latest report, in 2015, Washington State's greenhouse gas emissions were 97.4 million $tCO_{2eq}$, 7.4 million $tCO_{2eq}$ higher than the 1990 baseline of 90 million $tCO_{2eq}$.

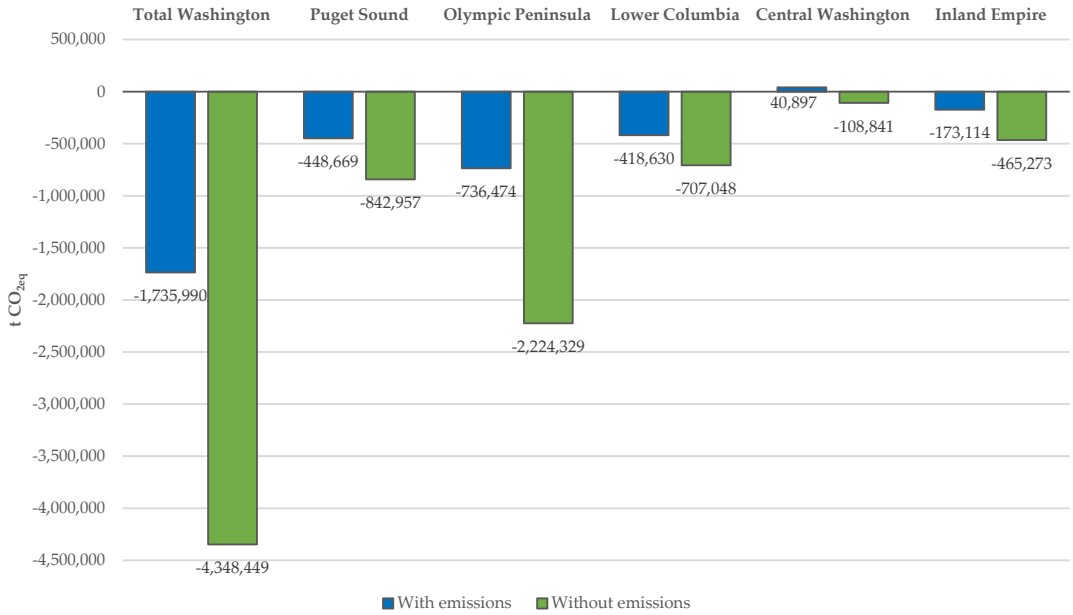

**Figure 5.** Global warming mitigating role of wood products in 2015 from private forests in Washington State. Results include the scenario where fossil fuel-based life cycle emissions from wood products manufacturing are included (blue) and excluded (green). The global warming potential is calculated over a 100-year timeframe.

The avoided emissions associated with the carbon storage of the Washington state's wood products output from private forests has a global warming mitigation potential equivalent to about 4.4% of the total state greenhouse gas emissions. If we account for the carbon sequestered by the net forest growth of private forests after harvesting, that corresponds to an additional 7.6%. Summing these two components, we can conclude that the total benefit on global warming of wood products and net forest growth (after harvesting) in private forests is equivalent to approximately 12% of the State's greenhouse gas emissions in 2015.

## 4. Discussion

This paper applies a radiative forcing analysis to Washington State's private forestlands, which supply the majority of the state's total forest harvest, and the state's wood products industry. We use 2015 harvest levels and product mix scenarios and demonstrate that these industries provide a global warming mitigation benefit, even after considering the emissions associated with forest harvest and product manufacturing.

The long-term climate benefit of harvested wood products calculated using the RF analysis and the 100-year average method are compared in Table 4. For each product, the 100-year average method is found to be conservative. The difference between the benefit of wood products calculated using the 100-year method and the RF analysis is 9.0% and 8.9% for lumber and plywood, respectively. Paper is found to be almost 41.2% less using the 100-year method because the RF method gives more weight to

storage in earlier years. This study validates that the 100-year average method roughly approximates the RF impact of temporary carbon storage.

**Table 4.** Long-term climate benefit (expressed per unit of product output in year 0) by primary product: a comparison using radiative forcing method and 100-year average method.

| Method | Lumber (kg) | Plywood (kg) | Paper (kg) |
|---|---|---|---|
| **Radiative Forcing (kgCO$_{2eq}$)** | 0.512 | 0.514 | 0.100 |
| **100-year average (kgCO$_{2eq}$)** | 0.466 | 0.468 | 0.059 |
| **Difference (kgCO$_{2eq}$)** | 9.0% | 8.9% | 41.2% |

When we apply the RF analysis to forest products, all products showed a climate benefit, except for the paper industry. However, the source of material for pulp derives from sawmill residues and clean harvest (and thinning) residues. In this case, it would be erroneous to consider the emissions associated with the paper industry in isolation as they fill an important market for the low-value co-products from sawn timber, and other wood products. Overall, while our analysis shows that on a purely product basis, Washington's paper industry does not provide a climate benefit. However, paper remains an important industrial product in the State and provides a market for the low-value co-product material of the other forest products industries.

Just as with product attribution, the various contributions of each landowner type are also difficult to look at in isolation. Wood product production shifted primarily to private lands in the 1990s after harvest on federal forests drastically reduced due to the listing of the northern spotted owl as an endangered species [58]. At a landscape level, Washington timberlands provide a mix of land and product sequestration, though individual hectares may contribute differently to each category.

*Limitations and Future Research*

This study did not consider the end of life scenarios for paper products, such as recycling. In the U.S., the recycling rate of paper and paperboard is very high, reaching 64.7% in 2014 [59]. Considering recycling scenarios may reduce the paper industry's climate impact, as recycling has been found to decrease the climate impacts of paper products [60,61]. In this analysis, the end of life of wood products is not included. Considering the alternative end of life scenarios for other forest products may enhance the overall climate benefits in this research. For example, a wood product that reaches its useful end of life may be burned in place of fossil fuels, thereby providing an additional climate benefit by displacing fossil fuel emissions [62]. Alternatively, if a product is sent to a landfill, the carbon would remain in a sequestered form for as long as the product remains in the landfill. Considering carbon stored in wood products in landfills has been shown to increase the carbon pool of multiple systems [63,64]. Further analysis could explore these alternative end of life scenarios and create a more complete picture of the global warming mitigation potential of different forest products.

Overall, this analysis may also understate the complete benefits associated with Washington's wood products because it does not consider the substitution effect. Substituting wood products in place of steel and concrete generally provides a climate benefit [65–67]. This is known as the substitution effect. There may be potential to increase the climate benefits associated with the industry by using more wood products and less steel and concrete products. Including the climate benefits of using wood products in place of alternative building materials in the analysis would likely improve the climate mitigation potential presented in this analysis and support the notion that wood products afford substantial climate benefits.

Moreover, in this study, we refrained from modeling forest management based 'what-if scenarios' and estimating the corresponding net environmental impacts of displacing non-wood products with wood-based products and vice versa. Scenario-based studies that fail to factor in the environmental effects of displacing non-wood products with bio-products (like wood) within their what-if scenarios, tend to propagate reduced harvest rates in certain forest types [68]. However, we know that the

substitution effect plays an essential role in the net environmental assessment of the wood products industry [69].

Forest management scenario analysis and the corresponding GHG reductions in the economy should factor in not only the growth rate of the forests but also the resultant wood products mix and the corresponding substitution effect. The modeling approach proposed in this paper provides the framework to address the wood products mix aspect of such what-if scenarios. However, this analysis does not consider alternative forest harvest scenarios but provides an estimate of the private timberlands and wood products industry in Washington State, based on the existing practice in 2015. It is therefore beyond the scope of this analysis to propose alternative forest management practices.

The aforementioned aspects can build on the results on the model presented in this paper. However, each of these aspects, not addressed in this paper, need attention to the specific nuances. Accordingly, the authors will address the aspects of recycling and substitution in a future publication.

## 5. Conclusions

The biogenic carbon sequestered in long-lasting wood products plays an important role in keeping the carbon away from the atmosphere for an extended period of time, providing an environmental service. Using the concept of temporal RF analysis, in this study, we estimate the impact of wood products on global warming, including carbon storage and life cycle GHG production/extraction emissions.

In this paper, we assess the role of Washington's wood products industry on the State's overall GHG emissions and the post-harvest carbon sequestration benefits. To develop a comprehensive estimate for the overall State, we used a combination of data sources. We used a moderate harvest rate scenario simulation within the Washington Biomass Calculator to estimate the yearly forest growth rate and harvest volumes from the States industrial and non-industrial private forestlands. We also used the Washington State Mill Survey data as a proxy for the wood products mix produced from harvested wood. Using the documented conversion rates, we estimated the volumes and fate of harvest and mill residues. The modeling results reveal that Washington State's wood products industry has a net mitigating effect on global warming. We estimate this temporal carbon storage lead to a global warming impact mitigation benefit equivalent to 4.3 million $tCO_{2eq}$. After factoring in the GHG emissions, associated with the harvest and manufacturing, within the proposed temporal model, the results produce a net beneficial impact of approximately 1.7 million $tCO_{2eq}$, making the wood products manufacturing industry in the State a net global warming mitigator.

Comparing these results with Washington State's annual greenhouse gas emissions, we can affirm that, overall, the global warming mitigation potential of Washington State's wood products industry from private forests is about 4.4% of the total greenhouse gas emissions (without considering the production emissions) in 2015. Including the net forest growth of private forests after harvesting, the global warming mitigation potential is equivalent to approximately 12% of the total greenhouse gas emissions.

Future research may, therefore, focus on (1) applying various end-of-life/recycling scenarios in an analysis of global warming mitigation potential; (2) incorporating the substitution effect to develop a comprehensive understanding of the net beneficial impacts of WA forest products industry; (3) evaluating the environmental benefits of extending lifetimes of forest products (like CLT), taking into consideration scenarios which would have the greatest climate benefit; (4) applying this method to different forest growth rates, harvest patterns, and product mix scenarios, carbon market scenario and compare relative impacts.

**Author Contributions:** Conceptualization, I.G. and F.P.; methodology, I.G. and F.P.; validation, I.G., F.P. and E.S.H.; formal analysis, I.G. and F.P.; writing—original draft preparation, I.G. and F.P.; writing—review and editing, I.G., F.P. and E.S.H.; visualization, I.G. and F.P.; supervision, I.G. All authors have read and agreed to the published version of the manuscript.

**Funding:** This research was funded by Washington State's "2019–2021-biennium operating funds" to CINTRAFOR and a gift from the Washington Forest Protection Association (WFPA).

**Acknowledgments:** The authors would like to acknowledge Jeff Comnick and Luke Rogers of the School of Environmental and Forest Sciences of the University of Washington, for providing the data for the total aboveground and harvest biomass in Washington State. Olivia Jacobs worked on developing the layered GIS maps and sections of the literature review. The authors would also like to thank Mark Doumit, Cindy Mitchell, John H. Ehrenreich, and Jason Callahan from the Washington Forest Protection Association (WFPA) for providing the industry insights.

**Conflicts of Interest:** The authors declare no conflict of interest. The funders had no role in the design of the study; in the collection, analyses, or interpretation of data; in the writing of the manuscript, or in the decision to publish the results.

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
