# Peer review of "Global Warming Mitigating Role of Wood Products from Washington State’s Private Forests"

_forests, doi:10.3390/f11020194_

Round 1

Reviewer 1 Report

Dear Authors,

Thanks for this study and resulting paper describing climate change mitigating effects of forests and wood products assessed at local level.

Please find here below suggestions from review process:

r17, r23 - it should be clearly stated to what these benefits of 4.3/7.4 mil. CO2eq, are related - management of ## hectares of private forests over period of ## years assuming that of ## cubic meters yearly increment ## cubic meters remain in forests yearly/over period of ## years accumulating standing volume (growing from recent ## cubic meters per hectare to ## after ## years) and ## cubic meters is extracted and processed to produce product mix characteristic for the region, all these figures should be also provided in a comprehensive table in Materials and methods section (information in table 2 is not sufficient in this context).

r25 - 12% of what reference year or for what period?

Abstract and Conclusions - if results are based on simulations of expected development, it should be clearly stated also in these sections.

r307-318 seems that this part comes from instructions for authors - should be removed.

r343-344 - more information on these adopted sustainable forest management practices would be appreciated - is it intentional to extract much less than the increment and what is the motivation for that - environmental goals?

r383-387 - numbers are somehow confusing when compared with those in Figure 2, please consider to add another graph.

r389 - is should be clearly stated, to products produced in what period results are related (a reference year or 5 years production or production in what period (as stated in Figure 5 caption - of wood products produced in 2015)).

r410-413 - for the figure on paper a brief interpretation of 100 years and more lasting emissions against short term effect of storage should be added.

r448 and also earlier in the text - is term "working forests" enough common?, maybe forests available/managed for wood production?

r452 ...wood products produced in 2015 contribute ...

r468-471 - assuming that in 1990 the effect of forest management and wood products production was similar than in 2015, there is no additional effect to achievements in other sectors now.

r477 - if it is on year 2015 products, then it is rather realistic product mix assessment than scenario.

Table 3 - units for rows a and columns should be added

Reviewer 2 Report

This is an interesting piece which needs some consideration of a number of concerns outlined below.

Acronyms are over-used in the manuscript. Forests is not an LCA journal and therefore acronym use should be minimized for the general forestry-focused reader.

LCA studies are replete with assumptions. The reader is left with a series of what-if questions. Ideally, this manuscript would a set of sensitivity analyses around the various assumptions showing how the results would differ based on variability in assumed values.

Again, thinking of Forests as a general category publication, extremely clear reasons for assumptions are warranted. Taking lumber as an example, it isn’t totally clear what different uses are assumed, and what the “lifetime” within each of those uses is assumed to be.

One can easily include that assumptions chosen and internal values/biases can lead to different interpretations of reality. Addressing other work with different interpretations would increase the value of the manuscript (e.g., Hudiburg et al. 2019).

Hudiburg, T. W., Law, B. E., Moomaw, W. R., Harmon, M. E., & Stenzel, J. E. (2019). Meeting GHG reduction targets requires accounting for all forest sector emissions. Environmental Research Letters, 14(9), 095005.

What about consideration of forest fires and land use change?

Lines 220-221 – “…for plywood…emission value between plywood…???

Lines 307-318 – that this text remains in the submitted version is not reassuring regarding the attention to detail in the complicated methods of the study.

Figure 1 is of insufficient resolution.

Table 2 – bone try tons, green tons, or?

Line 374 – what waste is referred to regarding paper production?

Figure 2 – source?

Figure 4 – why should the reader care about “without emissions”

Lines 428-429 - …carbon stored in the biomass of many wood products would further enhance the carbon sequestration capabilities of…forest products…???

Lines 432-433 – isn’t it obvious that extending product life enhances climate benefits?

Lines 471-473 – that this text remains in the submitted version is not reassuring regarding the attention to detail in the complicated methods of the study.

Round 2

Reviewer 2 Report

The authors have responded sufficiently to my comments. Some copy editing is needed in the text that was added based on reviewer comments.

Author Response

Reviewer Notes: The authors have responded sufficiently to my comments. Some copy editing is needed in the text that was added based on reviewer comments.

Authors' response: As was recommended by the reviewer we have copy-edited the entire document with special attention to the sections that were added based on the reviewer comments. 

The marked-up version of the manuscript is attached.
